# The Role of mTOR Signaling in Tumor-Induced Alterations to Neuronal Function in Diffusely Infiltrating Glioma

**DOI:** 10.3390/biomedicines13112593

**Published:** 2025-10-23

**Authors:** Hannah Haile, Sandra Leskinen, Arjun R. Adapa, Alexander R. Goldberg, Ashwin Viswanathan, Charlotte Milligan, Karen Conboy, Catherine Schevon, Peter Canoll, Brian J. A. Gill

**Affiliations:** 1Department of Neurological Surgery, Columbia University Medical Center, New York, NY 10032, USA; 2Department of Neurology, Columbia University Medical Center, New York, NY 10032, USA; 3Department of Pathology and Cell Biology, Columbia University Medical Center, New York, NY 10032, USA

**Keywords:** cancer neuroscience, glioblastoma, glioma, mTOR inhibitors, mTOR signaling, neuron–glioma crosstalk, peritumoral hyperexcitability, tumor microenvironment, seizure pathophysiology, synaptic remodeling

## Abstract

The mammalian target of rapamycin (mTOR) is a serine/threonine kinase that integrates metabolic and environmental signals to regulate cell growth and survival. In the central nervous system, mTOR plays a pivotal role in neuronal development, plasticity, and circuit homeostasis. In diffusely infiltrating gliomas, including glioblastomas, mTOR signaling is frequently dysregulated and contributes to malignant progression, therapeutic resistance, and metabolic adaptation. Beyond tumor-intrinsic effects, recent evidence reveals that gliomas actively reprogram peritumoral neurons via mTOR-dependent mechanisms, leading to synaptic remodeling, hyperexcitability, and neurological symptoms such as seizures and cognitive dysfunction. These results position mTOR as a central mediator of both oncogenesis and neurological dysfunction in diffusely infiltrating glioma. While clinical trials of mTOR inhibitors in gliomas have so far shown limited efficacy, emerging data suggest these agents may ameliorate tumor-associated neurological dysfunction. This review synthesizes current knowledge of mTOR signaling across tumor and neuronal compartments in diffusely infiltrating glioma and highlights its potential as a therapeutic target at the intersection of cancer biology and neuroscience.

## 1. Introduction

The mammalian target of rapamycin (mTOR) is a serine/threonine kinase that integrates extracellular and intracellular signals to modulate growth, protein synthesis, gene expression, and metabolic homeostasis [1,2]. While mTOR’s role in normal physiology is critical, its dysregulation is also implicated in various pathological states, including cancer, metabolic disorders, and neurodegenerative diseases [3,4,5,6].

Within the central nervous system, mTOR plays a role in regulating neuronal function, circuit maintenance, and neuroplasticity [7]. In the context of diffusely infiltrating gliomas (glioma) and other brain tumors, mTOR signaling influences both tumor biology and the surrounding neuronal environment. Tumor cells leverage mTOR signaling to promote proliferation, metabolic reprogramming, and resistance to therapy [8,9,10]. Peritumoral neurons undergo structural and functional changes—mediated in part by mTOR-driven alterations—that contribute to neurological dysfunction [11]. Thus, mTOR signaling has a dual role in the glioma microenvironment, where it is implicated in both tumor progression and neurological dysfunction.

This review aims to provide a comprehensive overview of mTOR’s contributions to glioma biology and tumor-induced neurological dysfunction. We will examine the molecular underpinnings of mTOR signaling in tumor cells and peritumoral neurons, emphasizing how these insights can inform the development of novel therapeutic strategies. By synthesizing current knowledge, this review seeks to establish a framework for leveraging mTOR-targeted therapies within the emerging field of cancer neuroscience, with the goal of improving oncological and neurological outcomes for patients with glioma and related conditions.

## 2. mTOR Signaling Network: Structure, Activators, and Downstream Mediators

As a member of the PI3K-related kinase family, mTOR functions as two distinct multiprotein complexes: mTOR Complex 1 (mTORC1) and mTOR Complex 2 (mTORC2) [12]. These complexes differ in their regulatory components and downstream effects, allowing mTOR to perform context-dependent roles in both normal and pathological cellular processes.

mTORC1 is primarily responsible for coordinating anabolic processes such as protein synthesis, lipid metabolism, and nucleotide biogenesis [13]. The activity of mTOR is regulated by upstream signals such as insulin, IGF1, and brain-derived neurotrophic factor which stimulate the PI3K-Akt pathway [14,15,16]. Akt phosphorylation inhibits the TSC1/TSC2 complex, a key negative regulator of mTORC1, allowing Rheb to activate mTORC1 on the lysosomal surface [17,18]. Nutrient availability, particularly of amino acids such as leucine and arginine, is sensed by the Rag GTPases, which recruit mTORC1 to the lysosome, enabling its activation by Rheb [19,20]. Conversely, energy depletion activates AMPK, which phosphorylates TSC2 and Raptor to suppress mTORC1, while hypoxia and oxidative stress inhibit mTOR signaling through the REDD1–TSC pathway [17,21,22,23]. These regulatory events occur at distinct phosphorylation sites, such that Akt-mediated phosphorylation inactivates TSC1/2, while AMPK-mediated phosphorylation enhances its activity, leading to opposing effects on mTORC1 signaling [17,22]. The activation of mTORC2 is less dependent on nutrient availability and is primarily driven by growth factors [24,25,26]. In both complexes, upstream PI3K-Akt signaling plays a central regulatory role: it activates mTORC1 indirectly by inhibiting the TSC1/2 complex and it activates mTORC2 directly, positioning PI3K as a key integrator of metabolic and growth signals [13,17,24,26] (Figure 1).

Once activated, mTOR signaling exerts its effects through a variety of downstream targets. mTORC1 promotes protein synthesis by phosphorylating the translational regulators 4E-BP1 and p70S6K, enhancing cap-dependent translation and ribosome biogenesis [27,28,29]. mTORC1 also regulates lipid metabolism via SREBP1/2-mediated transcription of lipogenic genes and suppresses autophagy by phosphorylating ULK1 and ATG13 [30,31,32,33,34,35]. mTORC2 contributes to cytoskeletal remodeling and cell survival by phosphorylating key signaling proteins such as Akt, SGK1, and PKC, enabling cells to respond to environmental changes and maintain structural integrity [36,37,38]. Together, the distinct yet interconnected functions of mTORC1 and mTORC2 highlight the complexity of this signaling network and the need for precise therapeutic strategies to modulate its activity in disease-specific contexts (Figure 1).

**Figure 1 biomedicines-13-02593-f001:**
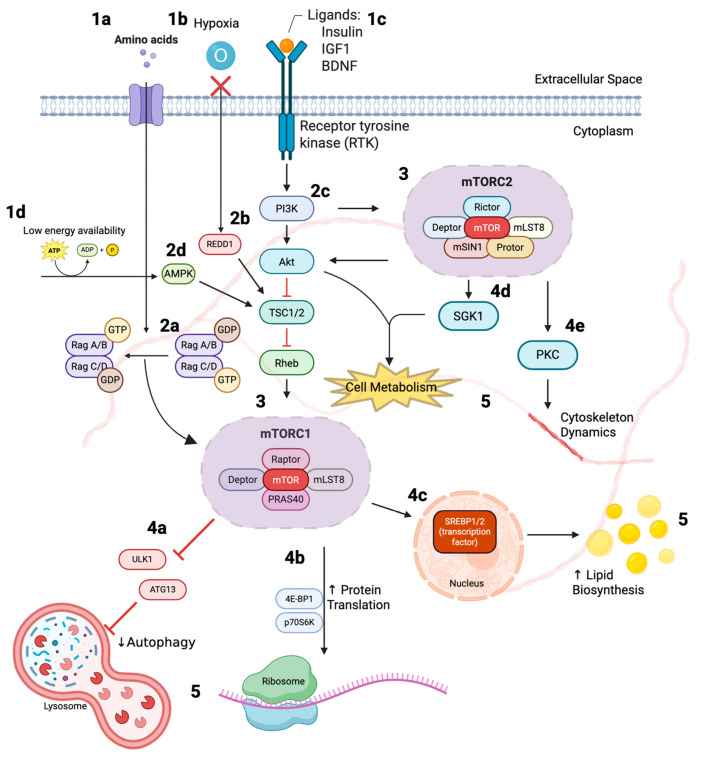
mTOR Signaling Pathways: (1) Upstream inputs include amino acids (1a), hypoxia (1b), growth factors such as insulin, IGF1, and BDNF (1c), and low energy availability (1d). (2) These cues are sensed and integrated through Rag GTPases (2a), REDD1 signaling (2b), PI3K–Akt signaling (2c), and AMPK signaling (2d) which converge on the TSC1/2–Rheb axis to regulate mTOR activity. (3) mTORC1 and mTORC2 are then activated at distinct cellular sites. (4) Mechanistically, mTORC1 suppresses autophagy (4a), promotes protein synthesis via p70S6K and 4E-BP1 phosphorylation (4b), and enhances lipid biosynthesis through SREBP1/2 (4c), while mTORC2 promotes cell metabolism through SGK1 (4d) and cytoskeletal remodeling via PKC (4e). (5) The integrated outputs of these pathways include increased protein translation, lipid biosynthesis, and cytoskeletal dynamics, alongside suppression of autophagy, thereby supporting tumor cell growth, proliferation, and survival [39]. Abbreviations: AMPK, AMP-activated protein kinase; Akt, protein kinase B; ATG13, autophagy-related protein 13; BDNF, brain-derived neurotrophic factor; Deptor, DEP-domain containing mTOR-interacting protein; EGFR, epidermal growth factor receptor; IGF1, insulin-like growth factor 1; mLST8, mammalian lethal with SEC13 protein 8; mSIN1, mammalian stress-activated protein kinase-interacting protein 1; mTOR, mechanistic target of rapamycin; mTORC1/2, mechanistic target of rapamycin complex 1/2; PI3K, phosphoinositide 3-kinase; PKC, protein kinase C; PRAS40, proline-rich Akt substrate of 40 kDa; Protor, protein observed with Rictor; Rag, Ras-related GTP-binding protein; Raptor, Regulatory-associated protein of mTOR; REDD1, regulated in development and DNA damage response 1; Rheb, Ras homolog enriched in brain; Rictor, rapamycin-insensitive companion of mTOR; SGK1, serum/glucocorticoid regulated kinase 1; SREBP1/2, sterol regulatory element-binding protein 1/2; TSC1/2, tuberous sclerosis complex 1/2; ULK1, Unc-51-like autophagy activating kinase 1; 4E-BP1, eukaryotic translation initiation factor 4E-binding protein 1; p70S6K, ribosomal protein S6 kinase.

## 3. mTOR Signaling in Neuronal Function and Development

The mTOR signaling pathway is indispensable for the development and maintenance of the central nervous system, orchestrating processes such as neuronal stem cell (NSC) proliferation and differentiation, neuronal morphology, and synaptic plasticity. mTOR signaling regulates the balance between NSCs self-renewal and differentiation, processes that are critical for brain development [40]. Loss of mTOR activity in a mouse model led to catastrophic depletion of NSCs, resulting in severe developmental abnormalities [41]. Conversely, hyperactivation of mTOR can prematurely exhaust stem cell niches, leading to apoptosis of stem cell progenitors, thus reducing the pool of proliferating NSCs and disrupting the timing of differentiation, compromising long-term neurogenic potential and leading to abnormal brain architecture [42,43,44,45]. These disruptions underlie several neurodevelopmental disorders, including tuberous sclerosis complex (TSC), focal cortical dysplasia (FCD), and hemimegaloencephaly (HME), which arise from either germline or somatic mutations in mTOR pathway components (Table 1) [22,43,46,47,48,49].

Beyond its role in NSC regulation, mTOR signaling shapes neuronal morphology, connectivity, and excitability of post-mitotic neurons [40,50,51,52]. In developing neurons, mTORC1 promotes dendritic arborization and axon elongation by modulating local protein synthesis and cytoskeletal dynamics, processes critical for the formation of functional neural circuits [43,53,54,55,56]. mTORC2, through its regulation of the actin cytoskeleton, influences neuronal polarity and dendritic spine morphogenesis [52,57,58,59,60]. In addition, mTORC2 promotes neuronal survival and excitability by phosphorylating Akt, PKC, and SGK1, linked with broader metabolic activity and electrophysiological functions [36,37]. Crosstalk between mTORC1 via mTORC2 activity occurs via Akt, which suppresses TSC1/2 and thereby promotes mTORC1 activation, while mTORC1 in turn exerts negative feedback on PI3K signaling that can attenuate both mTORC1 and mTORC2 activity [17,24]. In mature neurons, mTOR supports synaptic plasticity by controlling activity-dependent translation of synaptic proteins, thus linking extracellular signals to long-lasting changes in circuit function [50,54,55].

**Table 1 biomedicines-13-02593-t001:** Summary of mTORopathies.

Disorder/Pathology	Gene/Pathway Alteration	mTOR Signaling State	Key Features
Tuberous Sclerosis Complex	TSC1, TSC2 mutations	Hyperactivated	Cortical tubers, epilepsy, ASD, cognitive deficits, SEGA [7,16,22,42,43,46,61]
PTEN Hamartoma Tumor Syndrome	PTEN mutation/deletion	Hyperactivated	Macrocephaly, epilepsy, neuronal hypertrophy, ASD-like behavior [40,43,62,63,64,65,66,67]
Hemimegaloencephaly	Somatic mTOR/PI3K/AKT3 mut.	Hyperactivated	Unilateral brain overgrowth, seizures, cortical dysplasia [48,49,68,69,70]
Focal Cortical Dysplasia	Somatic mTOR pathway mut.	Hyperactivated	Cortical thickening, abnormal lamination, epilepsy [47,48,49,68,69,70,71]
Age-associated cognitive decline	Decreased mTOR signaling	Hypoactivated	Impaired neurogenesis, reduced NSC proliferation, memory deficits [40,72,73,74,75,76]
Neurodegeneration (various forms)	mTOR dysregulation	Variable	Inclusion bodies, synaptic loss, microglial activation [40,72,73,74,75,76]

## 4. Pathologies of Altered mTOR Signaling in the Brain

Aberrant mTOR signaling is a hallmark of several neurological disorders—collectively termed “mTORopathies”—including genetic syndromes, cortical malformations, and brain tumors. Most involve hyperactivation of mTORC1, though reduced activity can also impair NSC function and contribute to neurological dysfunction [7,40,43,77].

TSC, the archetypal mTORopathy, is caused by mutations in TSC1 or TSC2—negative regulators of mTORC1—resulting in constitutive activation of the pathway. This dysregulation leads to the formation of hamartomas—including cortical tubers and subependymal nodules—and subependymal giant cell astrocytomas (SEGA) (Table 1) [22,43,46,61]. Neurologic manifestations include epilepsy, intellectual disability, and autism spectrum disorder (ASD) (Table 1) [7,17,42,43]. In murine models, loss of TSC disrupts axon guidance, dendritic spine morphology, and synaptic plasticity, leading to seizures through impaired autophagy [7,78,79,80,81]. Inhibition of mTORC1 reverses many of these neurological phenotypes, linking mTOR hyperactivation to circuit dysfunction in TSC [7,11,13,40,43,81].

Loss-of-function mutations in the tumor suppressor PTEN similarly lead to unchecked activation of the PI3K-AKT-mTOR axis (Table 1) [82,83,84]. PTEN deletions lead to neuronal hypertrophy and increased synaptic excitation via mTORC1 signaling and are strongly associated with neurodevelopmental disorders including macrocephaly, ASD and epilepsy (Table 1) [40,43,62,63,64,65,66,67]. These effects are rescued by mTORC1 inhibition [66,85].

FCD and HME result from somatic mutations in mTOR pathway genes, including mTOR, PI3K, and AKT3 (Table 1) [48,49]. These mutations result in mosaic hyperactivation of mTORC1, leading to cortical enlargement, neuronal misplacement, and seizures (Table 1) [68,69,70]. The pathological overlap between cortical tubers in TSC and lesions seen in FCD reflects a shared mTOR-driven pathogenesis [47,71].

Although hyperactivation of mTOR is most implicated in neuropathology, inadequate mTOR signaling can also be detrimental. Hypoactivation of mTOR—whether age-related or experimentally induced—impairs NSC maintenance and neurogenesis, contributing to age-associated cognitive decline (Table 1) [40,72,73,74,75,76].

Collectively, these examples illustrate how dysregulated mTOR signaling contributes to a diverse array of neuropathological processes, ranging from neurodevelopmental to neurodegenerative disorders.

## 5. mTOR in Gliomas

Diffusely infiltrating gliomas (gliomas) are the most common primary brain tumors in adults, encompassing a heterogeneous group of tumors which are classified based on their histologic and molecular features [86]. Characteristic molecular alterations include IDH1/2 mutations, TP53 mutations and ATRX loss in astrocytomas; IDH1/2 mutations and chromosome 1p/19q codeletions in oligodendroglioma; and TERT promoter mutations, EGFR amplification and chromosome 7 gain/10 loss in glioblastoma (GBM) [86]. Prognosis varies markedly by molecular subtype. IDH1/2 mutant gliomas—particularly those with more favorable histologic features—are associated with a greater survival than IDH1/2 wildtype GBM which is characterized by a median survival of approximately 15 months [87,88,89,90,91,92,93]. Additionally, gliomas are associated with high rates of neurologic morbidity such as seizures, cognitive decline, and focal deficits [94,95,96,97,98,99].

Aberrant mTOR activation is a hallmark of gliomas and is driven by upstream alterations such as EGFR amplification, PTEN loss, and PI3K mutations, all converging to increase signaling through the PI3K-Akt-mTOR pathway [100,101,102,103]. Elevated mTORC1 activity promotes anabolic metabolism, protein synthesis, and angiogenesis within the tumor microenvironment, thereby supporting glioma cell proliferation and survival under stress conditions such as hypoxia and nutrient deprivation [12,104,105,106].

In glioma stem-like cells, mTORC1 activity supports self-renewal and therapy resistance, and dual inhibition of mTORC1/2 reduces tumorgenicity and disrupts cytoskeletal dynamics critical for glioma invasion [9,107,108]. Moreover, inhibition of mTOR signaling can sensitize glioma cells to genotoxic stress, highlighting its relevance as a therapeutic target in both treatment-resistant and recurrent disease [109,110,111,112].

Collectively, these findings underscore the multifaceted role of mTOR signaling in glioma biology, encompassing tumor growth, progression, and therapeutic resistance.

## 6. mTOR-Dependent Mechanisms Underlying Tumor-Induced Neurological Dysfunction in Glioma

Emerging evidence has revealed that gliomas reprogram peritumoral neurons through mTOR-dependent mechanisms, contributing to tumor-induced neurological dysfunction [11,113,114,115,116].

In murine glioma models, peritumoral neurons exhibit mTOR hyperactivation as evidenced by increased phosphorylation of ribosomal protein S6. Neuron-specific ribosomal profiling and patch-clamp electrophysiology studies in murine glioma models, demonstrate that neuronal mTOR hyperactivation is implicated in the translational downregulation of both excitatory and inhibitory post-synaptic proteins, loss of dendritic spines, an elevated resting membrane potential, and depolarizing responses to GABAergic stimulation in excitatory neurons [11,113]. Altered chloride homeostasis may also contribute to depolarizing GABAergic responses in peritumoral neurons. PI3K/Akt signaling has been shown to phosphorylate the chloride co-transporters NKCC1 via WNK3 and KCC2, shifting intracellular chloride gradients and thereby modifying the net effect of GABA from inhibitory to depolarizing [117,118,119,120]. Tumor-induced hyperactivation of neuronal mTOR also affects the function of peritumoral inhibitory neurons, decreasing their firing rate and altering the timing of their firing with respect to excitatory neuron activity [113]. Collectively these alterations to the translatome, morphology and electrophysiological properties of excitatory and inhibitory neurons promote a more excitable state in peritumoral tissue. Remarkably, these alterations are fully reversed within hours of acute treatment with the ATP-competitive mTORC1/2 kinase inhibitor AZD8055, demonstrating the direct and dynamic role of mTOR in mediating the tumor-induced alterations to peritumoral neurons [11,113] (Figure 2).

Tumor-induced mTOR-driven alterations to peritumoral neurons contribute directly to the cortical dysfunction observed in peritumoral tissue. In vivo EEG recordings demonstrated an increase in spontaneous epileptiform activity during tumor progression [113]. Furthermore, in vivo two-photon calcium imaging of excitatory peritumoral neurons in functional cortex demonstrated the presence of tumor-induced alterations to physiological stimulation [11]. Specifically, peritumoral neurons in the somatosensory cortex exhibited increased response amplitude and delayed timing to whisker stimulation [11]. Importantly, acute mTOR inhibition with AZD8055 reduced epileptiform activity and normalizes the excitatory neuron response to physiological stimulation in functional cortex, demonstrating its ability to reverse tumor-induced neurological dysfunction in vivo [11,113]. Altogether, these findings establish mTOR as a key mediator of tumor-induced neurological dysfunction and highlight its translational potential as a target for restoring circuit level function in brain tumors (Figure 2).

## 7. Clinical Applications of mTOR Inhibitors

Despite considerable preclinical interest in targeting the mTOR pathway in glioma, clinical trials using mTOR inhibitors have demonstrated limited efficacy in slowing tumor progression [122,123,124,125]. However, the efficacy of mTOR inhibition as a treatment for tumor-induced neurological dysfunction in humans remains unexplored.

Rapamycin analogs (rapalogs) such as everolimus and temsirolimus were among the first mTOR inhibitors evaluated in GBM patients. These agents selectively inhibit mTORC1 through FKBP12 binding, but do not suppress mTORC2, resulting in incomplete inhibition of the mTOR pathway. This has been shown to trigger compensatory activation of upstream pathways such as PI3K/AKT signaling, which may contribute to therapeutic resistance [126,127,128]. Moreover, rapalogs demonstrate variable blood–brain barrier penetration and are associated with systemic immunosuppression, limiting their dosing and duration in clinical settings [110,129]. Clinical trials of everolimus (NCT01062399; NCT00387400) and temsirolimus (NCT01019434; NCT00022724) in GBM and high-grade glioma have shown modest anti-tumoral activity but have fallen short of delivering a significant survival benefit, while also not evaluating the potential effects on glioma-associated seizures or other neurological complications [126,130,131,132,133,134,135]. One explanation for these limitations is that selective mTORC1 inhibition does not block upstream PI3K activity, which continues to drive both mTORC1 and mTORC2 signaling. Targeting the PI3K-Akt-mTOR axis more broadly may therefore represent a more effective therapeutic approach [100,127].

In response to the limitations highlighted by first-generation rapalogs, second generation ATP-competitive inhibitors were developed that effectively inhibit both mTORC1 and mTORC2 and block the feedback activation of upstream PI3K/AKT signaling. The dual inhibitors AZD8055, Sapanisertib (INK128), and Torin-1 achieve more comprehensive mTOR inhibition and exhibit antitumor efficacy in preclinical glioma models [127,129]. However, these agents appear largely cytostatic rather than cytotoxic and have yet to demonstrate significant anti-tumoral benefit in glioma patients [110,128]. Early-phase trials have reported dose-limiting toxicities (NCT02619864), and no dual mTOR inhibitors have yet progressed to routine clinical use in neuro-oncology [126].

mTOR inhibition has shown significant clinical success in non-malignant neurological dysfunction, suggesting they may be therapeutic in combating tumor-induced neurological dysfunction. The clearest example is TSC, marked by constitutive mTOR activation due to mutations in either TSC1 or TSC2 [136]. In this setting, everolimus has received FDA and EMA approval for the treatment of SEGA and for treatment-refractory epilepsy, with heightened focus on pediatric uses [137,138,139]. Clinical trials have demonstrated significant reductions in seizure frequency, tumor volume, and disease progression, shown in the EXIST 3 study (NCT01713946) [140,141,142,143]. These findings confirm that mTOR pathway modulation can influence neural network excitability and seizure thresholds when hyperactivation is genetically driven. Beyond TSC, mTOR inhibitors are also under investigation for mTOR-related epilepsies such as FCD, another mTORopathy, where early data suggest improved seizure control and slowed disease progression [142,144].

Building on these findings mTOR inhibitors may offer a novel therapeutic avenue in glioma—not solely for tumor suppression, but for mitigating the tumor-induced neurological dysfunction seen consistently in both preclinical models and in clinical care [11,145]. Whether such neuromodulatory benefits can be achieved in tumor-induced neurological dysfunction remains an important and unanswered question. Despite promising preclinical evidence, no clinical trials have directly evaluated neurologic outcomes such as seizure control, cognition, or neurological deficits in glioma patients receiving mTOR inhibitors [11,110,113,141].

A major challenge limiting the use of mTOR inhibitors is their systemic toxicity, which can include metabolic disturbances, pulmonary complications, and myelosuppression [146]. To mitigate these risks, local delivery strategies are under investigation. Biodegradable polymers such as acetalated dextran (Ace-DEX) scaffolds have co-delivered everolimus and paclitaxel with tunable release kinetics, expanding on the precedent of Gliadel wafers [147,148]. Convection-enhanced delivery offers homogeneous intraparenchymal drug distribution, bypassing blood–brain barrier limitations [149,150]. Preclinical studies using polymers and lipid nanoparticles have achieved focal rapamycin delivery with improved pharmacokinetics and reduced systemic exposure [126,127]. Such approaches may be particularly valuable for addressing peritumoral hyperexcitability, where systemic dosing may be ineffective or intolerable.

## 8. Discussion and Future Perspectives

mTOR signaling occupies a unique intersection between oncogenesis and neurobiology, serving as a critical regulator of cell growth, metabolic adaptation, and synaptic function. In the context of glioma, mTOR not only sustains tumor growth intrinsically but also orchestrates complex and dynamic interactions between glioma cells and adjacent neuronal networks. This review synthesizes the evidence underscoring the multifaceted roles of mTOR as both a driver of tumor progression and mediator of tumor-induced neuronal dysfunction, thus highlighting its position as a crucial molecular conduit linking tumor progression with neurological morbidity.

Despite increasing preclinical support, the clinical translation of mTOR inhibitors for tumor-induced neurological dysfunction remains limited. Clinical trials investigating mTOR inhibitors in glioma have not explicitly assessed neurological outcomes such as seizure frequency or cognitive function. This gap highlights the importance of incorporating more specific neurological endpoints to fully determine the clinical benefits of targeting mTOR signaling.

Moving forward, research should emphasize several critical directions to fully realize the therapeutic potential of targeting mTOR in glioma. First, clinical trials must integrate specific neurological outcomes, including seizure burden, cognitive performance, and patient-reported quality of life, to capture the benefits of mTOR inhibition more accurately. Second, the development of innovative delivery systems such as convection-enhanced delivery, nanoparticle encapsulation, or biodegradable implants could substantially improve the precision and effectiveness of mTOR inhibitors by enabling localized modulation of signaling at the infiltrative margin. Third, combination strategies targeting complementary mechanisms involved in neuron–tumor crosstalk, such as synaptic plasticity, immune modulation, and metabolic reprogramming, should be rigorously evaluated. Finally, advanced preclinical models, including patient-derived organoids and brain slice cultures, are important for better recapitulating neuron–glioma interactions and accelerating the translation of promising therapeutic strategies into clinical practice. Ultimately, embracing these multifaceted approaches may facilitate breakthroughs in managing both tumor progression and neurological symptoms, significantly enhancing outcomes for patients with glioma (Figure 3).

## Figures and Tables

**Figure 2 biomedicines-13-02593-f002:**
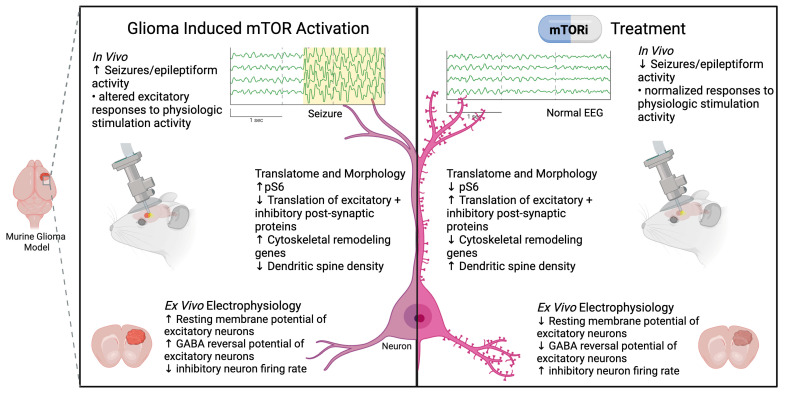
mTOR-mediated Neuronal Dysfunction in Glioma: Tumor-induced mTOR activation alters peritumoral neuronal function across multiple levels, including increased phosphorylated ribosomal protein S6 (pS6), reduced synaptic protein translation, and dendritic spine loss; electrophysiologic changes such as elevated resting membrane potential and impaired inhibitory signaling ex vivo; and seizure activity with aberrant excitatory responses in vivo. Treatment with mTOR inhibitors reverses these effects, normalizing translatome profiles, stabilizing membrane properties, and suppressing seizure activity [121].

**Figure 3 biomedicines-13-02593-f003:**
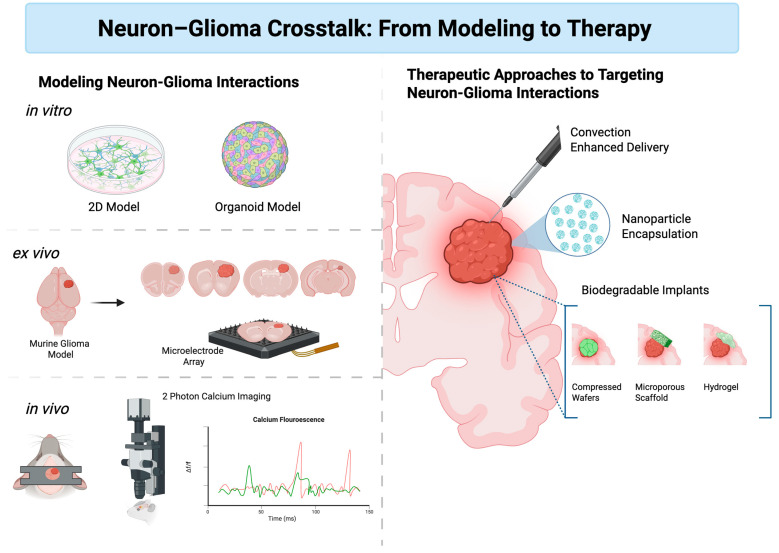
Neuron–Glioma Crosstalk: From Modeling to Therapy. On the left, modeling platforms span in vitro systems such as 2D neuronal–glioma co-cultures and iPSC-derived organoids, ex vivo acute brain slices combined with microelectrode array recordings, and in vivo murine glioma models incorporating techniques such as two-photon calcium imaging with schematic traces depicting green fluorescence in glioma cells and red fluorescence in neurons. On the right, novel therapeutic approaches are shown including convection-enhanced delivery of agents targeting aberrant signaling, nanoparticle encapsulation for improved pharmacokinetics, and biodegradable implants such as compressed wafers, microporous scaffolds, and hydrogels for local drug release. Together, these approaches highlight how experimental models of neuron–glioma signaling can be tested and then implemented using novel treatment modalities [151].

## Data Availability

Not applicable.

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
