# Peer review of "The Role of mTOR Signaling in Tumor-Induced Alterations to Neuronal Function in Diffusely Infiltrating Glioma"

_biomedicines, 2025, doi:10.3390/biomedicines13112593_

Round 1

Reviewer 1 Report

Comments and Suggestions for Authors

This review article focused on the role of mTOR in glioma growth and infiltration. It contains scientific valuable information and good figures. However, if authors add more information in some part it will turn the review more descriptive for clinical and drug development.

  • Phosphorylation of TSC1/2 by two kinasees AKT and Ampk has two distinct effect on activation/inhibition of TSC1/2 which may related to the distinct phosphorylation sites in the TSC1/2. I suggest that these point add to the section mTOR signaling.
  • - What is the role of mTORc2 beside it's effects on cytoskeletal system in neuronal cells? Is there any cross talk between them?
  • It seems one of the most key regulator of mTORC1/2 activator is PI3K which activate mTORC1 and mTORC2. PI3k activity orchestrate developmental neuronal activity. Moreover failure of mTORC1 inhibitors in clinical trials in the section of clinical applications of these drugs exactly support my suggestion. Thus  focus on PI3K role may lead to better drug discovery in the glioma therapy field.
  • One important point on GABAergic depolarizing function may related to the role of chloride co-transporter function by activation of PI3k/AKT which may phosphorylated both NKCC1 and KCC2 transporter and change the physiological effects of GABA. It will be worth to mention this in the text.

Author Response

Thank you very much for taking the time to review our manuscript titled “The Role of mTOR Signaling in Tumor-Induced Alterations to Neuronal Function in Diffusely Infiltrating Glioma”. Please find the detailed responses below and the corresponding revisions/corrections highlighted/in track changes in the re-submitted files.

Comment 1: Phosphorylation of TSC1/2 by two kinases AKT and Ampk has two distinct effect on activation/inhibition of TSC1/2 which may related to the distinct phosphorylation sites in the TSC1/2. I suggest that these point add to the section mTOR signaling.

Response 1: We thank the reviewer for this comment. We have revised the manuscript to add this point (lines 71-73 clean and tracked).

Comment 2: What is the role of mTORC2 beside its effects on cytoskeletal system in neuronal cells? Is there any crosstalk between them?

Response 2: We thank the reviewer for this comment. We have included more information on the role of mTORC2 in neurons and the crosstalk between mTORC1 and mTORC2 in the manuscript (lines 161 – 167 tracked; lines 133 – 139 clean).

Comment 3: It seems one of the most key regulator of mTORC1/2 activator is PI3K which activate mTORC1 and mTORC2. PI3k activity orchestrate developmental neuronal activity. Moreover, failure of mTORC1 inhibitors in clinical trials in the section of clinical applications of these drugs exactly support my suggestion. Thus, focus on PI3K role may lead to better drug discovery in the glioma therapy field.

Response 3: In the mTOR Signaling Network section, we now clarify that PI3K activity regulates both mTORC1 and mTORC2, positioning PI3K as a central integrator of metabolic and neuronal signaling pathways (lines 75-78 clean and tracked). In the Clinical Applications section, we also note that the limited efficacy of selective mTORC1 inhibitors in glioma may be explained by persistent PI3K activity, which continues to activate both complexes, and that targeting the broader PI3K–Akt–mTOR axis may represent a more effective therapeutic approach (lines 289 – 293 tracked; lines 261-265 clean).

Comment 4: One important point on GABAergic depolarizing function may related to the role of chloride co-transporter function by activation of PI3k/AKT which may phosphorylated both NKCC1 and KCC2 transporter and change the physiological effects of GABA. It will be worth to mention this in the text.

Response 4: In the section on tumor-induced neurological dysfunction, we have added a note that PI3K/Akt signaling can phosphorylate the chloride co-transporters NKCC1 and KCC2, altering chloride gradients and contributing to depolarizing GABA responses in peritumoral neurons (lines 239 – 243 tracked; lines 211 – 215 clean).

Reviewer 2 Report

Comments and Suggestions for Authors

The study by Haile et al. critically evaluates mTOR’s contributions to glioma biology and tumor-induced neurological dysfunction. Overall, the manuscript makes a favorable impression and fits the scope of the Biomedicines journal.

However, in my point of view, the current version has certain shortcomings that may diminish reader engagement.

Please find below my suggestions that may improve the manuscript.

Major critiques:

  1. Figure 1. It would be very helpful if the authors indicated numbers on the figure to help follow the main stages/sequence of the reaction cascade. A detailed description of each numbered step should be provided in the Figure Legend. In the legend, please also provide the full names for the abbreviations shown in the figure.
  2. I suggest expanding Section 5 by starting with a brief overview of gliomas to prepare the reader for the subsequent information.
  3. It would be helpful if the authors indicated the ID numbers of the clinical trials discussed in Section 7.
  4. It would be beneficial if the authors could find a way to graphically illustrate the ideas presented in Section 8 (Discussion and Future Perspectives), for example, by creating a summary figure or schematic.

There are some minor typos and stylistic inaccuracies; please double-check the text.

Author Response

Thank you very much for taking the time to review our manuscript titled “The Role of mTOR Signaling in Tumor-Induced Alterations to Neuronal Function in Diffusely Infiltrating Glioma”. Please find the detailed responses below and the corresponding revisions/corrections highlighted/in track changes in the re-submitted files.

Comment 1: It would be very helpful if the authors indicated numbers on the figure to help follow the main stages/sequence of the reaction cascade. A detailed description of each numbered step should be provided in the Figure Legend. In the legend, please also provide the full names for the abbreviations shown in the figure.

Response 1: We thank the reviewer for this comment. We have updated Figure 1 to include a numbered sequence of events, and the figure legend to reflect the numbers with a full list of abbreviations used in the figure (lines 90 – 113 clean and tracked).

Comment 2: I suggest expanding Section 5 by starting with a brief overview of gliomas to prepare the reader for the subsequent information.

Response 2: We thank the reviewer for this comment. In Section 5, we have added a brief overview of gliomas to provide context for the subsequent discussion. Specifically, we now include a summary of glioma classification, their prognosis and associated neurologic morbidity, along with key molecular alterations that define clinically relevant subgroups (lines 204 – 214 tracked; lines 176 – 186 clean).

Comment 3: It would be helpful if the authors indicated the ID numbers of the clinical trials discussed in Section 7.

Response 3: We have included study ID numbers where applicable in Section 7 (lines 285 – 286, 301, 309 – 310 tracked; lines 257 – 258, 273, and 281 – 282 clean).

Comment 4: It would be beneficial if the authors could find a way to graphically illustrate the ideas presented in Section 8 (Discussion and Future Perspectives), for example, by creating a summary figure or schematic.

Response 4: We included Figure 3, a schematic of modeling and therapeutic approaches to neuron-glioma interactions, in the Discussion and Future Perspectives (lines 365 – 376 tracked; lines 336 – 347 clean).

Round 2

Reviewer 2 Report

Comments and Suggestions for Authors

The authors have addressed all my comments. I have no further concerns.